# OpenReview forum: "Reproducibility Study: Understanding multi-agent LLM cooperation in the GovSim framework"
_TMLR — Accepted by TMLR_

### Review · Reviewer_XAjX · 2025-03-21

**Summary Of Contributions:**

The work reproduces and extends the Governance of the Commons Simulation framework, challenging the original claims from several aspects.
- The work challenges the claim in the original paper that only large LLMs achieve sustainable cooperation. It finds that state-of-the-art small models (DeepSeek-R1-Distill) can sustain equilibrium, suggesting improvements in fine-tuning enhance cooperative abilities.
- The study confirms that universalization-based reasoning improves sustainability, but finds that the effect stems more from numerical instructions than abstract moral reasoning. It also expands GovSim by introducing seven new social reasoning frameworks (e.g., utilitarianism, deontology, Rawls’ maximin principle) and adding human-agent interaction support, broadening its applications.

**Audience:**

Yes

**Claims And Evidence:**

No

**Requested Changes:**

My concerns regarding the contributions of this paper are as follows:
1. The original paper does not seem to place that much emphasis on model size in comparison to the capabilities of the model, which aligns with the claims presented in this work. It would benefit from a more systematic analysis of the specific aspects of fine-tuning that contribute to the observed performance improvements. Such an analysis could help clarify the factors driving the enhancements in the model’s effectiveness.
2. The study posits that universalization-based reasoning is predominantly effective due to numerical cues rather than ethical considerations. However, the paper does not fully address or eliminate other potential contributing factors, such as improved strategic planning or implicit reasoning patterns. To substantiate this claim, further controlled experiments that isolate these variables would provide stronger empirical support and add rigor to the argument.
3. In the appendix, the authors report runtime for models of varying sizes. Did the authors compare the run time of larger vs. distilled smaller models in GovSim?

Overall, I find the analysis presented in this paper to be underdeveloped. A deeper, more comprehensive analysis is needed to provide a clearer understanding of the original work.

**Strengths And Weaknesses:**

Strengths:
- The paper is clearly written.
- The study refines the impact of universalization-based reasoning, demonstrating that explicit numerical instructions drive sustainability improvements more than abstract ethical reasoning.


Weakness:
- The study claims that fine-tuning improves cooperation but does not explicitly quantify the mechanisms enabling smaller models to perform well, which could benefit from a deeper analysis.
- The impact of human-agent interaction is introduced but not explored in depth.

---

### Review · Reviewer_Y3gJ · 2025-04-21

**Summary Of Contributions:**

This is a reproducibility study focusing on the GovSim framework. Reproducing the baseline experiments unveils discrepancies in some of the considered metrics compared to the original paper (maybe due to stochasticity), hinting some caution. New results include evaluating the performance of (a) recent smaller distilled models and (b) additional social reasoning methods on GovSim.

**Audience:**

Yes

**Claims And Evidence:**

Yes

**Requested Changes:**

- I feel section 3.1.5 is kinda disconnected to the rest of the paper. Is this feature used in the expert's advice test? If yes I would suggest to explicitly mention
- I found parts related to the impact of universalization to sub-skills difficults to follow. I am referring to both the setup/motivation of this experiment as well as its results. Could the authors elaborate more on this?
- It would be useful if the authors could provide insights behind the following finding connecting the nuances of the different environments in GovSim to the effect of different social reasoning strategies
>  Lastly, we observed that the effect of the social reasoning strategies appear to be highly dependent on the scenario, with large discrepancies between scenarios for the same models.

**Strengths And Weaknesses:**

Strengths:

- Clean write-up with well-explained motivation and goals.
- I found the social reasoning evaluation interesting and the results therein insightful.
- Evaluating/integrating DeepSeek models in GovSim is of practical use
- Raising issues of modularity in GovSim

Weaknesses:

- The paper's contributions are rather limited, albeit interesting.
- Some contributions of the paper were not clear to me, which I think can be attributed to an interconnection issue among subsections.
- The analysis and interpretation of some of the findings was lacking.

See requested changes below for more details.

---

### Review · Reviewer_csFz · 2025-04-27

**Summary Of Contributions:**

This submission presents a reproducibility study of the Governance of the Commons Simulation (GovSim), which models cooperation and sustainability between LLM agents in resource-sharing scenarios. The study replicates the original experiments and extends the framework by testing smaller models, such as DeepSeek-R1, which were found capable of achieving sustainable outcomes, challenging the original claim that only large models could. It also confirms that universalization-based reasoning improves agent sustainability, although the improvement is likely due to numerical instructions rather than the reasoning principle itself. The study introduces seven additional social reasoning frameworks, revealing that reasoning with numerical prompts significantly enhances cooperation. Additionally, it extends the framework to allow human-AI interactions, offering new insights into mixed-agent collaboration. The paper contributes to the understanding of LLM-based multi-agent systems and ethical AI, providing a foundation for future research in cooperation, scalability, and human-AI collaboration.

**Audience:**

Yes

**Broader Impact Concerns:**

No immediate broader impact concerns.

**Claims And Evidence:**

Yes

**Requested Changes:**

- The study should address the discrepancies in the over-usage and equality metrics. These discrepancies could be due to the stochastic nature of resource allocation or differences in metric definitions. It is essential to clarify how these metrics are calculated and to ensure that they are consistent across all experiments. A more transparent explanation of the metric calculation methodology would strengthen the validity of the results and ensure reproducibility. This is critical for the acceptance of the paper, as inconsistencies in key metrics could undermine the credibility of the findings.
- The paper suggests that improvements in agent performance using universalization-based reasoning may primarily stem from the numerical instructions provided, rather than the reasoning framework itself. To strengthen this claim, the authors should investigate this in more depth by testing other reasoning frameworks without numerical guidance and comparing the results. A clearer distinction between the effects of reasoning principles and the role of numerical instructions would strengthen the paper’s contributions. This is critical for securing the recommendation for acceptance, as it influences the theoretical implications of the study.
- Due to computational constraints, the study uses a smaller subset of models than the original work. While the results with smaller models like DeepSeek-R1 are promising, it would be beneficial to expand testing to include a broader range of models. This would help validate the findings more robustly across a wider spectrum of LLMs and better reflect the diversity of current model architectures. The authors could consider including a broader set of models, particularly closed-source models, to strengthen the study. While this would strengthen the work, it is not critical for acceptance.

**Strengths And Weaknesses:**

# Strengths
- The paper does an excellent job of replicating the original experiments from the Governance of the Commons Simulation (GovSim) framework, ensuring that the findings are reproducible. It provides a detailed methodology, making it easy for others to verify and build upon the results.
- A key strength is the demonstration that smaller LLMs, such as DeepSeek-R1, are now capable of achieving sustainable equilibrium in resource-sharing simulations. This challenges the original assumption that only large models could succeed, offering new insights into the advances in small-model capabilities.
- The paper extends GovSim by integrating seven new social reasoning frameworks, such as utilitarianism and virtue ethics, into the simulations. This broadens the scope of the research and enhances its relevance to ethical AI and decision-making.

# Weaknesses
- While the study replicates the original experiments, there are some discrepancies in the metrics, particularly in the over-usage and equality metrics. These inconsistencies could be clarified, and it would help to ensure that these metrics are defined and implemented consistently across studies.
- The paper suggests that the improvement from universalization-based reasoning is largely due to the numerical instructions given to the agents. While this is insightful, it raises the question of whether the framework's focus on reasoning principles is sufficiently developed, or if the improvement is more about the directiveness of the instructions themselves. Further investigation into this aspect could strengthen the claim.
- Due to computational constraints, the study was limited to a smaller subset of models compared to the original study. While this is understandable, it may be beneficial to expand the testing to include a broader variety of models, especially considering that advances in LLMs are rapidly evolving.

---

### Review · Reviewer_53nb · 2025-04-27

**Summary Of Contributions:**

The paper replicates the GovSim framework (Piatti et al., 2024), focusing particularly on small open-sourced LLMs. They replicate the core experiments (baseline, universalization, sub-skills tests) using a subset of the original small open-source LLMs (Llama-2, Llama-3, Mistral) and extend the study by including recent DeepSeek R1 distilled variant models. Beyond reproduction, the paper introduces and tests seven additional social reasoning frameworks and extends the GovSim codebase to allow for human participation. Key findings include confirming that universalization improves sustainability, showing that recent small reasoning models can achieve sustainable equilibria, unlike the older small models tested originally. They also provide evidence suggesting the performance improvement from universalization stems more from the explicit numerical guidance in the prompt than the abstract reasoning principle itself.

**Audience:**

No

**Claims And Evidence:**

No

**Requested Changes:**

- (Important) Better presentation of contributions beyond replication (for human interaction feature in 3.1.5, if the authors decide to keep it, it should be bolstered by a convincing analysis and clear presentation; for others, better presentation of results and findings)
- Deepseek itself is a reasoning model, which is fundamentally different than LLaMA and mistral, which the authors frame as the smaller models in the original paper. The reasoning nature of deepseek and its training inevitably creates a confounding effect, perhaps compounded by the "social reasoning" interventions in the paper. The authors didn't address this point in the manuscript, it would be helpful to in the minimum examine the reasoning trace by the deepseek distilled model and see whether the reasoning goes in line or in opposite directions to the reasoning interventions originally proposed by Piatti et al., 2024.
- (optional) More experiments, see earlier section.

**Strengths And Weaknesses:**

Strength:

- Tests with deepseek distilled models show improvement
- Clearly written

Weaknesses:

- The very short section on human interaction feature is not explained clearly in the paper
- The replication of the original paper are limited (only open sourced, small models)
- Insights from the newly introduced approach limited: the point on human participation, as written above, is poorly supported; the point on subskill and social reasoning in Figure 2 and Figure 3 seem to lack statistical rigor (Figure 2: only 4 points on the graph, Figure 3: most points stuck on 20)
- "A temperature of 1.0 led these models to excessively long responses, leading to simulation issues." - Can the authors clarify what simulation issues they are referring to and why this couldn't be addressed.
- The paper uses distilled deepseek on llama and qwen, in line with its focus on small models, but this seems odd given that the actual Deepseek R1 haven't been tested anywhere before on this framework. According to the findings I would expect it to do much better, but there's no actual experiments on this.
- Small models as limiting scope - small models is somewhat interesting, but with the compute resources now and widely available access to off the shelf LLMs, only focusing on small LLMs (that ran within 40G GPU memory) seems limiting, with the wide array of sometimes very cheap (and cheaper than A100 GPU) inference services available, that will give much better performance. In this aspect, it's not entirely clear the interests it will provide to the audience of TMLR.

---

### Review · Reviewer_FxrH · 2025-05-27

**Summary Of Contributions:**

The paper investigates the properties of a Large Language Model (LLM) multi-agent framework that is designed to study cooperation and sustainability between LLM agents in a resource-sharing environment.  The particular framework is called Governance of the Commons Simulation (GovSim).  In particular the authors perform simulation experiments in order to reproduce the findings of the original GovSim paper (Piatti et al., 2024).  One of the findings of the original paper was that small LLMs fail to achieve a sustainable equilibrium in a resource-sharing environment.  However, one of the findings of this paper hints to the contrary; i.e., recent small LLMs (in particular, DeepSeek-R1-Distill-Qwen-14B) are indeed able to achieve a sustainable equilibrium.  Furthermore, the authors perform experiments using different approaches of reasoning strategies, thus indicating that the behavior of an agent is highly correlated to the specific prompt that they receive for a particular task.

In what follows I will describe my point of view for the particular paper.  However, please note that my remarks come with low confidence as I have not done any research in this field and moreover I have no interest in doing any work with LLMs in the near future.  This is a review of a paper that the authors submitted on May 21, 2025.  I have not reviewed an earlier version of this paper.

**Audience:**

Yes

**Broader Impact Concerns:**

No concerns from my end.

**Claims And Evidence:**

No

**Requested Changes:**

Please see the typos and other weaknesses as listed above.

**Strengths And Weaknesses:**

**Strengths**

**S1.**  The paper is largely easy to read.  This is true even for someone who is a complete outsider in this field like myself.

**S2.**  Organizing the information in the tables given was good.  Furthermore, I appreciate that more information is available in the appendix.  However, I have admittedly did not look into these details.


**Weaknesses**

**W1.**  While the paper is largely easy to read, in some cases I was wondering what was what.  For example, in the first paragraph of page 2 the authors discuss universalization-based reasoning.  However, in all honesty, while I understand what the authors want to say at a high-level, I have no idea about anything specific.  Not to mention what "universalization" means at that point.  So, if they wouldn't mind giving an explanation in a brief paragraph in the appendix, this might be good.

**W2.** Perhaps add an outline for the paper at the end of Section 1?  Also, a link to your repository?  In the following page you have a footnote with a link to the repository of the original GovSim paper.  However, you have modified/extended that code.  So, why not make your code public?  I think this is really counting against you in terms of claim and evidence presented in your work.

**W3.** Section 4.1, lines 6-7: "smaller models can now also survive 12 months without depleting the resource" -> This was nowhere to be found that it was a goal in the study.  It seems reasonable that this is the goal in the original paper regarding resource-sharing, but I do not believe this has been explained at that point in the paper so far.

**W4.** Captions of tables.  Tables 1 and 2 (page 6) have the captions below the tables.  Table 3 in page 8 has the caption above the table.  Please see the guidelines of TMLR and be consistent.

**W5.** In Figure 3 I was a little bit puzzled with the choice of colors. We are looking at the mean survival time and I presume that the larger values are better, where I presume that the values of the heatmap correspond to months in simulation time.  Along these lines, the largest value appears to be red and I think this is counter-intuitive because usually red is used to increase someone's caution or show something prohibitive.  So, I would either change the colors of the heat-map, or add a sentence in the caption explaining a little bit better what we are looking at.

**W6.** Some typos here and there:
- page 2, second enumerated item: "...powerful LLM ever reach..." -> "...powerful LLMs ever reach..." (plural for LLM, or "reach" should be "reaches")
- page 3, Section 3.1.2 starts subskills with lower-case S (contrasting Section 4.2 which uses capital S)
- page 4, 4 lines before the end of page: decision -> decisions (plural)
- page 4, 3-4 lines before the end of the page: the reference should be listed as (Dallimore & Mickel, 2011)
- page 5, Section 3.1.4, paragraph 2, line 2: table 1 -> Table 1 (capital "T")
- page 5, Section 3.2, line 7: appendix D -> Appendix D (capital "A")

---

### Decision · Action_Editor_Ucak · 2025-06-12

**Recommendation:** Accept with minor revision

**Additional Comments:**

1. Statement on “Numerical Guidance”

- The manuscript asserts that the primary reason universalisation reasoning improves model performance is the explicit “numerical guidance” in the prompts, rather than the abstract ethical principles themselves. This is an intriguing finding. However, several reviewers (Reviewer XAJX, Reviewer csFz) contend that the current experimental design—while providing correlational evidence—fails to fully rule out other confounding variables and therefore does not establish a convincing causal relationship.
- **Revision suggestion:** As the authors noted in their responses to Reviewer csFz and Reviewer XAJX, please supplement the paper with ablation experiments measuring the effect of universalisation on each sub‑skill.

2. Explanation of “Small Model Success”

- The paper challenges the notion that “only the largest models can succeed,” demonstrating that newer, smaller models (e.g., DeepSeek‑R1‑Distill‑Qwen‑14B) can also reach sustainable equilibria, which the authors attribute to advances in model architecture and instruction fine‑tuning. However, as requested by Reviewer XAJX, the manuscript does not provide a mechanistic analysis explaining why these fine‑tuning procedures lead to improved collaborative abilities.
- **Revision suggestion:** The authors have responded that such an analysis lies beyond the scope of the current work. It is recommended that they either replace this claim with more conservative wording or remove it entirely.

3. Claim Regarding “Human–Machine Interaction”

- Although the Introduction (Section 1) claims that the proposed framework supports human–machine interaction, the manuscript offers no empirical evidence or experiments to substantiate this assertion—mentioning it only again in the “Future Work” section.
- **Revision suggestion:** I recommend that the authors incorporate appropriate experiments in the final version to validate this aspect of the framework, or else remove the related discussion from the Introduction.

**Audience:**

Yes

**Audience Explanation:**

- Given TMLR’s scope of publishing experimental and theoretical studies that yield new insights into the behavior of learning systems—including reproducibility studies and new analytical frameworks—readers interested in rigorous benchmarking of emergent behaviors in large language models via the GovSim platform will find this work directly relevant.
- Furthermore, TMLR emphasizes sound empirical validation and technical correctness over subjective significance, making the clear, open‐source implementation and reproducible results presented here particularly valuable to its audience.

**Claims And Evidence:**

Yes

**Claims Explanation:**

- The authors demonstrate that **GovSim reliably enables the study and benchmarking of emergent sustainable behavior** by reproducing the original baseline experiments: their mean survival-time curves closely match those reported in the reference study (Figure 1), confirming that the platform yields consistent and interpretable metrics across scenarios.
- They challenge the original assertion that “only the largest and most powerful LLMs reach a sustainable outcome” by showing that **recent instruction-tuned, smaller open-weight models (e.g., DeepSeek-R1-Distill-Qwen-14B and Phi-4) now achieve full 12-month survival**, narrowing the gap with much larger models and reflecting advancements in model design and tuning.
- The study confirms that **universalization-based moral reasoning significantly improves cooperative sustainability**, with a paired right-tailed t-test revealing average gains of +1.83 months survival time, +8.19 total gain units, and +6.82 % efficiency (p = 0.04), thereby substantiating that this social reasoning prompt boosts long-term outcomes.
- By extending social‐reasoning experiments to include seven frameworks and three prompt instructiveness levels, the authors show that **prompt specificity—not the ethical principle alone—is the primary driver of improved cooperation**: prompts embedding explicit, scenario-specific calculations (“Univ. with calc”) consistently outperform abstract variants (Figure 3), and Level 3 prompts with worked numerical examples yield the highest efficiency gains (Figure 4).